# Phylogeographic reconstruction of the marbled crayfish origin

Julian Gutekunst[1], Olena Maiakovska[1], Katharina Hanna[1], Panagiotis Provataris[1], Hannes Horn[2], Stephan Wolf[2], Christopher E. Skelton[3], Nathan J. Dorn  [4,5✉] & Frank Lyko  [1✉]

The marbled crayfish (*Procambarus virginalis*) is a triploid and parthenogenetic freshwater crayfish species that has colonized diverse habitats around the world. Previous studies suggested that the clonal marbled crayfish population descended as recently as 25 years ago from a single specimen of *P. fallax*, the sexually reproducing parent species. However, the genetic, phylogeographic, and mechanistic origins of the species have remained enigmatic. We have now constructed a new genome assembly for *P. virginalis* to support a detailed phylogeographic analysis of the diploid parent species, *Procambarus fallax*. Our results strongly suggest that both parental haplotypes of *P. virginalis* were inherited from the Everglades subpopulation of *P. fallax*. Comprehensive whole-genome sequencing also detected triploid specimens in the same subpopulation, which either represent evolutionarily important intermediate genotypes or independent parthenogenetic lineages arising among the sexual parent population. Our findings thus clarify the geographic origin of the marbled crayfish and identify potential mechanisms of parthenogenetic speciation.

[1] Division of Epigenetics, DKFZ-ZMBH Alliance, German Cancer Research Center, 69120 Heidelberg, Germany. [2] Genomics and Proteomics Core Facility, German Cancer Research Center, 69120 Heidelberg, Germany. [3] Skelton Environmental Consulting, LLC, Madison, GA 30650, USA. [4] Department of Biological Sciences, Florida Atlantic University, Davie, FL 33314, USA. [5] Department of Biological Sciences, Florida International University, North Miami, FL 33181, USA. ✉email: ndorn@fiu.edu; f.lyko@dkfz.de

The marbled crayfish (*Procambarus virginalis*) is a novel parthenogenetically reproducing freshwater crayfish species s[1,2]. The animals were originally introduced into Germany in 1995, as an ornamental aquarium pet from the USA[2], and then became rapidly distributed through the aquarium trade[1,3]. Following anthropogenic releases, *P. virginalis* has now formed large colonies in diverse habitats across Europe[4,5]. In addition, the animals are widely distributed in the freshwater networks of Madagascar[6–8] where they are increasingly exploited as a source of nutritional protein for human consumption[9,10].

Microsatellite analyses suggested a genetically uniform pattern for all *P. virginalis* analyzed[11,12]. Furthermore, cytogenetic analyses uncovered a defective first meiotic division and a triploid karyotype[13,14]. These findings, in combination with the absence of detectable recombination[5] suggested that the animals reproduce by apomictic parthenogenesis, thus generating genetically uniform offspring. The recently completed first draft assembly of the *P. virginalis* genome[8] revealed a relatively large (3.5 Gb) genome with a triploid AAB genotype, consistent with earlier observations[14,15]. Comparative genome sequencing of numerous specimens from diverse sources and locations confirmed that the global *P. virginalis* population is largely monoclonal with only very limited genetic differences between geographically separated populations[2,5,8]. This monoclonality can be reasonably explained by a recent founder event involving a single animal. The identity of this foundational animal has remained largely unknown.

Based on morphological and genetic markers, the slough crayfish (*Procambarus fallax*) has been suggested as the parent species of *P. virginalis*[16]. *Procambarus fallax* are sexually reproducing diploids that are endemic to peninsular Florida and southern Georgia[17]. While *P. virginalis* appears to have diverged from *P. fallax* as recently as 25 years ago[2,15], the genetic, phylogeographic, and mechanistic origin of the species has remained unresolved, resulting in the perception that marbled crayfish are enigmatic "mutants". Based on genetic analyses, it has been suggested that *P. virginalis* may have arisen by a captive pairing of distantly related *P. fallax* parents, perhaps from different parts of the geographic range[8]. Another possibility is that *P. virginalis* represents a naturally occurring form of *P. fallax* that was isolated and propagated through the aquarium trade. Based on current knowledge, it is not possible to distinguish between these hypotheses and little effort has been made to detect *P. virginalis* in Florida. We have now constructed a new genome assembly for *P. virginalis* to support a detailed phylogeographic analysis of 92 *P. fallax* from 23 locations across the native range and to identify the putative parent population(s) of *P. virginalis*.

## Results

To clarify the origin of the marbled crayfish, we collected *P. fallax* specimens from 23 different locations covering the entire native range of the species (Tab. S1). Samples were numbered by their sampling locations from 1 (northernmost) to 23 (southernmost). For genotyping, we used partial sequences of the mitochondrial cytochrome b (CytB), nuclear DNA methyltransferase 1 (Dnmt1), and mitochondrial cytochrome c oxidase 1 (Cox1) genes (see Methods for details). Sequencing of PCR amplicons from up to three markers and from 92 specimens showed that every analyzed *P. fallax* had at least one polymorphism (Fig. 1A) with the published *P. virginalis* reference sequence[8]. These findings strongly suggest that *P. virginalis* is not present within the native range of *P. fallax*.

For a phylogeographic analysis of the *P. fallax* population, we selected one representative specimen from each location (numbered 1–23) for whole-genome sequencing (Tab. S2). For five locations, where the PCR results had suggested a greater level of genetic heterogeneity, two specimens were selected (labeled "a" and "b"). Sequencing reads were subsequently mapped on the marbled crayfish mitochondrial reference genome sequence to reveal maternal haplotypes. The results showed a substantial genetic diversity of the mitochondrial genomes and defined four populations (Fig. 1B). These populations were broadly aligned with the four major water catchment areas of Florida (Fig. S1) and were named correspondingly (Fig. 1B). Our results also showed that the monoclonal *P. virginalis* reference sequence was deeply nested in the Everglades population (Fig. 1C). In fact, *P. virginalis* formed a tight cluster with two specimens (19 and 21b) that were collected from independent locations (70 km apart) in the south-eastern Everglades.

For a detailed analysis of nuclear genome relationships, we generated an improved *P. virginalis* genome assembly by long-read sequencing with a total length of 3.7 Gb and a weighted mean sequence length (N50) of 144.4 Kb (Tab. S3). Compared with the previously published version[8], this assembly reduced the number of total scaffolds from more than 3 million to 170 thousand, and the number of unknown bases from 1663 million to 663 million (Tab. S3). Quality controls showed a negligible amount of bacterial contamination within the genome assembly (Fig. S2). Benchmarking with universally conserved single-copy orthologs[18] revealed 70.5% complete (3.4% duplicated), 11.8% fragmented, and only 17.7% missing orthologs, which is similar to other published arthropod genomes and further confirms the high quality of our assembly.

Mapping of the whole-genome sequencing datasets from the 28 *P. fallax* specimens (Tab. S2) further clarified the *P. fallax* populations that provided the parents of the foundational *P. virginalis* specimen. Principal component analysis revealed a population structure that appeared similar to the one supported by the analysis of mitochondrial genomes and again suggested a close relationship between *P. virginalis* and the Everglades population of *P. fallax* (Figs. 2, S3). While traditional haplotype analysis was hampered by the polyploid genome structure, we could identify the majority ("A") and minority ("B") haplotype for every heterozygous position in the *P. virginalis* nuclear genome and then define the sequence of these positions for every *P. fallax* genome. The resulting scores were subsequently used to analyze the genetic relationships between the *P. virginalis* haplotypes and the complete set of *P. fallax* nuclear genomes. The results showed a population structure that was similar to the structure observed with mitochondrial genomes, with both *P. virginalis* haplotypes clearly related to the Everglades population of *P. fallax* (Fig. S4). This analysis also confirmed the particularly close relationships of the two specimens with the most closely related mitochondrial DNA sequence (19 and 21b, Fig. S4).

We have previously used allele frequency distributions of heterozygous positions to demonstrate that the marbled crayfish genome is triploid[8]. Interestingly, the *P. fallax* specimens 19 and 21b, that were most closely related to *P. virginalis*, also showed clear allele frequency distribution peaks at 0.33 (Fig. 3A), which is similar to *P. virginalis* (Fig. 3A) and consistent with triploidy. In sharp contrast, all other *P. fallax* specimens showed allele frequency distribution peaks at 0.5 (Figs. 3A, S5), in agreement with a diploid genome. Similar to *P. virginalis*, the triploid *P. fallax* genomes showed a negligible fraction of triallelic sequence polymorphisms (Fig. 3B), suggesting that they recapitulate the AAB genotype of *P. virginalis*. However, both animals also showed a considerable number of genetic polymorphisms (>100,000 homozygous single nucleotide variants) towards the *P. virginalis* reference genome (Fig. 3C). This is more than an order of magnitude above what has been described for various other *P. virginalis* specimens[5] and demonstrates that these animals are genetically distinct from *P. virginalis*. Altogether our findings thus

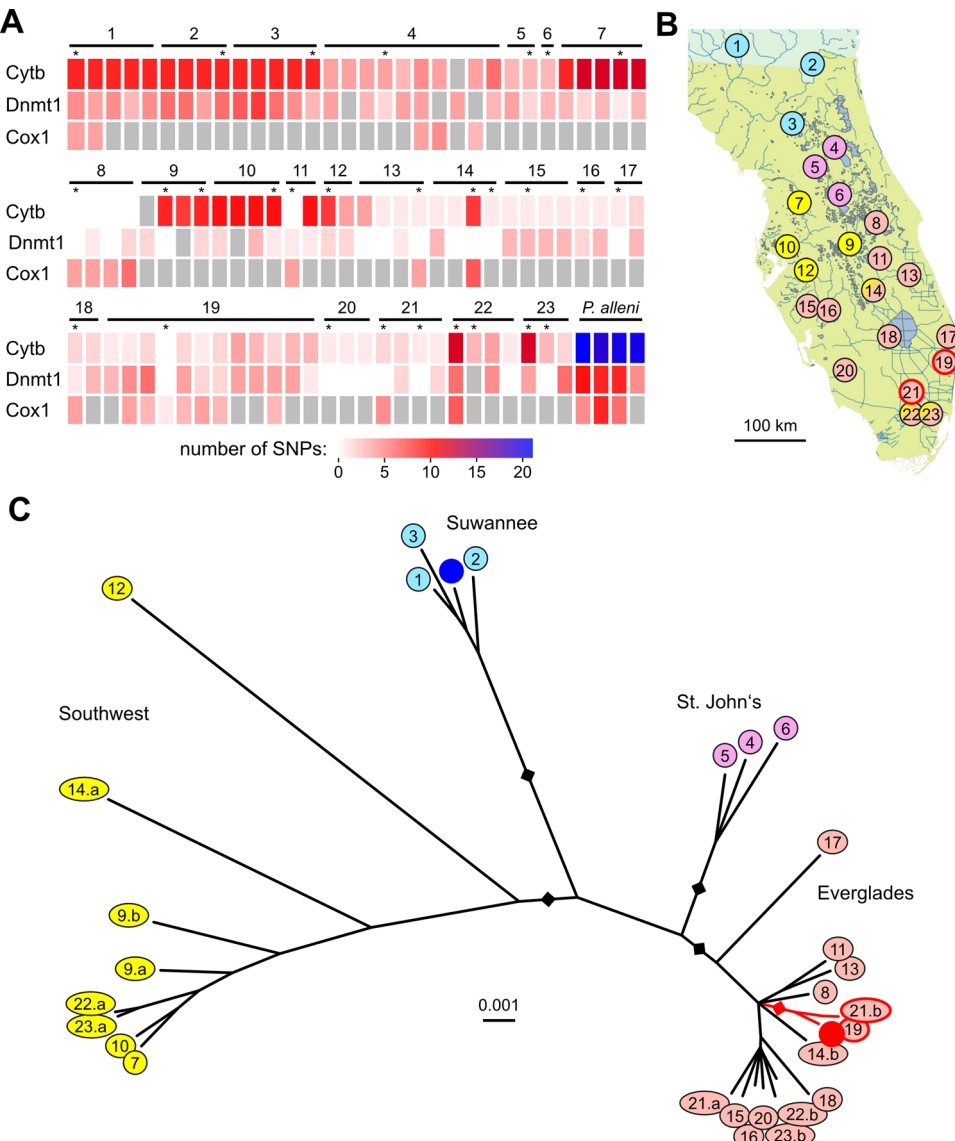

**Fig. 1 Phylo-geographic analysis of wild-caught *Procambarus fallax*. A** PCR genotyping results of 92 *P. fallax* sampled for this study. The heatmap indicates the number of genetic polymorphisms for specific markers relative to the *P. virginalis* reference, on a scale from 0 (white) to 21 (blue). Grey boxes: not analyzed, asterisks denote animals that were also analyzed by whole-genome sequencing. The morphologically similar and sympatric crayfish species *P. alleni* ($N = 4$) was included as a control and showed a clearly distinct genetic profile. Sample numbers indicate individual sampling locations from 1 (northernmost) to 23 (southernmost); a and b indicates independent animals collected from the same location. **B** Map of Florida and southern Georgia, indicating the collection sites and the population structure. Three locations showed local sympatry of the Southwest and Everglades populations, as indicated by split coloring. **C** Phylogenetic tree, based on complete mitochondrial genome alignments. Colors indicate phylogeographically defined populations that were named after the corresponding water catchment areas of Florida. The blue dot indicates the *P. fallax* mitochondrial genome reference sequence, the red dot indicates the monoclonal *P. virginalis* mitochondrial genome reference sequence. Animals that are particularly closely related to *P. virginalis* are highlighted by red lines. Rectangles represent bootstrap values for the branches that define subpopulations (black: 100%, red: 95%).

identify a subpopulation of *P. fallax* that shares important genetic features with *P. virginalis* and likely represents the foundational population of *P. virginalis*.

## Discussion
Our study allowed us to reconstruct the genetic makeup of the foundational *P. virginalis* specimen from the existing populations of *P. fallax*. Our findings strongly suggest that the marbled crayfish is a direct descendant of *P. fallax*, with both parental haplotypes inherited from the Everglades population, thus contradicting our previous hypothesis for a hybrid-like speciation event[8]. Our genotyping of 92 total specimens from Florida detected no evidence for the presence of *P. virginalis*. This is

consistent with a recent review of 2299 *P. fallax* museum specimens, which revealed a female ratio of 55.3%, similar to other freshwater crayfish species from Florida, and suggests the absence of "hidden" parthenogenetic populations[19]. Future work should focus on increasing the sampling across *P. fallax* populations in Florida, in order to more accurately determine the population frequency and geographic distribution of *P. fallax* triploids. Additional analyses will be required to determine the reproductive mode of triploids and to better understand their ecology.

While we did not detect any *P. virginalis* in Florida, we did identify triploid *P. fallax* in the Everglades population, which were highly similar, but not identical to *P. virginalis*. We currently do not know the reproductive mode of these animals. Because the

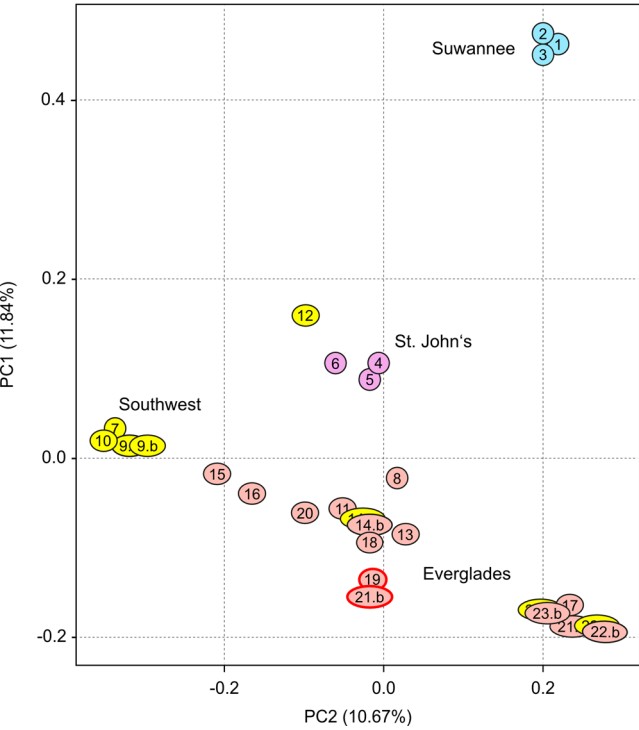

**Fig. 2 Principal component analysis, based on linkage-pruned variant sites (N = 76,863) obtained from the *Procambarus fallax* whole-genome sequencing dataset.** Colors indicate the four major subpopulations: Suwannee (blue), St. John's (purple), Southwest (yellow), and Everglades (red). Closely overlapping red/yellow ovals represent independent animals collected from the same location ("a" and "b"). Animals that are particularly closely related to *P. virginalis* are highlighted by red ovals.

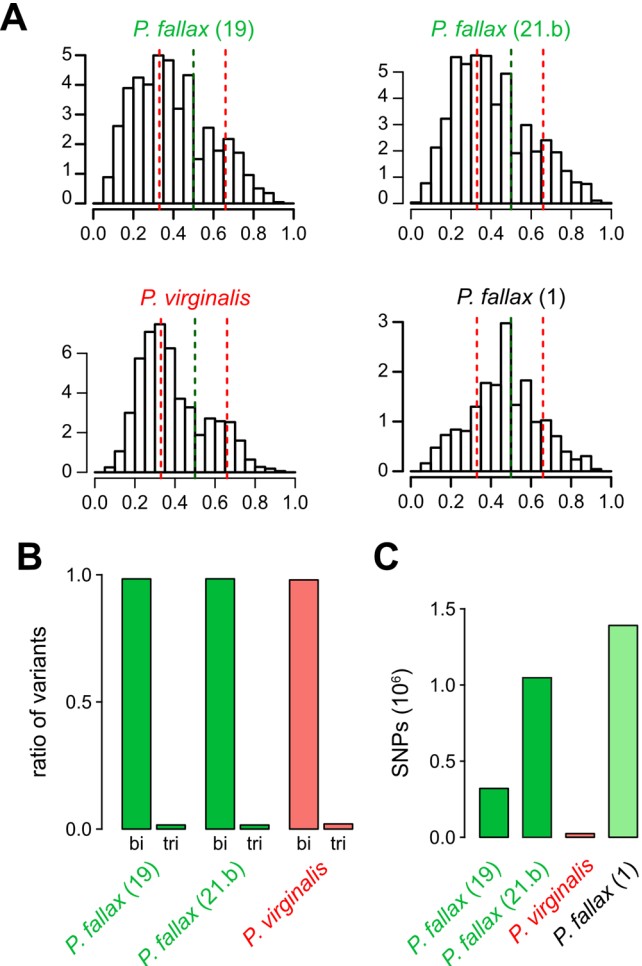

**Fig. 3 Triploidy in *Procambarus fallax*. A** Heterozygous allele frequency distribution in two triploid *P. fallax*, in *P. virginalis* and a diploid *P. fallax*. The histograms show the frequencies of heterozygous alternative allele ratios. Numbers for allele frequencies are indicated as counts (in $10^5$). **B** Distribution of biallelic (bi) and triallelic (tri) SNPs in *P. virginalis* and in triploid *P. fallax*. **C** Numbers of biallelic SNPs, relative to the *P. virginalis* reference genome.

combination of triploid and haploid gametes usually results in aneuploidy, abortive development, and/or sterility[20], we consider it unlikely that they could reproduce sexually. Thus, triploid *P. fallax* should be either sterile or parthenogenetic. Triploid parthenogenetic *P. fallax*, similar to marbled crayfish, would be expected to show superior fitness compared with sexually reproducing diploids[21] and should thus be frequently found in *P. fallax* populations. However, the presence of triploids among our samples was relatively rare, with only two out of 28 specimens, suggesting that these animals are likely sterile. In comparison, triploid parthenogenetic individuals in populations of the New Zealand mud snail (*Potamopyrgus antipodarum*) were substantially more frequent (53%, *N* = 860)[22]. It is possible that parthenogenetic triploid *P. fallax* occasionally arises, but is disfavored by environmental[23] or competitive[24] factors. These factors may be absent in the non-native habitats into which *P. virginalis* has been released through anthropogenic activities.

## Methods

***Procambarus fallax* collections and PCR genotyping.** Animals were collected from various wild populations (Table S1) in compliance with state and local regulations (Georgia department of natural resources scientific collection permit 115621108, state of Florida collection permits S-19-10 and S-20-04). DNA was isolated from abdominal muscle tissue using SDS-based extraction and precipitation with isopropanol. PCR genotyping of 92 specimens was performed using the following primer pairs: Cytb (FWD CAGGACGTGCTCCGATTCATG and REV GACCCAGATAACTTCATCCCAG), Dnmt1 (FWD GCTTTCTGGTCTCGTAT GGTG and REV CTGCACACAGCCTAAGATGC), Cox1 (FWD CTGCTATTG CTCATGCAGGT and REV TGCCCGAGTATCTACATCCA). Amplicons were verified by agarose gel electrophoresis and cloned using the TOPO TA Cloning Kit (Invitrogen) according to the manufacturer's instructions. The purified plasmids were sequenced by eurofins genomics and the sequences were aligned using SnapGene software.

***Procambarus virginalis* PacBio genome sequencing and assembly.** Genomic DNA was isolated from three independent animals using SDS-based extraction and precipitation with isopropanol. PacBio large-insert library preparation was done following the recommended protocols by Pacific Biosciences. For each animal, a library was generated from 1 to 5 µg of sheared and concentrated DNA with an insert size target of ~10 kb. After library preparation, sequencing was performed on a PacBio SEQUEL platform according to the manufacturer's instructions. Movie times were 600 for most SMRT cells with some being 240, 360, and 900. Sequencing results for animals were pooled resulting in a total of 37 SMRT cells comprising 69,074,290 reads and 242,146,920,794 bases.

Long subreads generated from the PacBio SEQUEL platform were assembled into contigs using the Canu assembler[25] version 1.7. After self-error correction and trimming for reading quality estimates and SMRT cell adapters, the remaining 31,652,687 reads, comprising 91,697,556,862 bp, were used in the assembly phase. Computations were done on a high-performance cluster running the Slurm Workload Manager using up to 56 CPUs and 450 GB memory. To achieve an improved gene representation, contigs were connected using transcriptome information. Briefly, all transcripts were mapped onto the contig assembly and linkage information was extracted to connect contigs using L_RNA_SCAFFOL DER[26] version 1.0. Contiguity of sequences was improved using proximity ligation based on Chicago and Hi-C methods (provided by Dovetail Genomics, Chicago, USA) using fresh frozen tissue of one additional animal. After DNA extraction and library preparation, libraries were on an Illumina HiSeq X platform using a PCR-free paired-end 150 bp sequencing protocol. Reads were processed for scaffolding and error-correcting using Dovetail's HiRise scaffolding pipeline (Dovetail Genomics, Chicago, USA).

Automatic annotation was performed as previously described[8]. Briefly, sequences larger or equal 10 kb were extracted from the assembly and annotated using the MAKER pipeline[27] version 3.00. Annotation data were provided by the manually curated Uniprot/Swiss-Prot database[28] and the annotated marbled crayfish transcriptome[8]. Functional domains were predicted using InterproScan[29] version 5.39–77.0.

In order to assess the assembly quality, a defined set of single-copy ortholog arthropod genes were searched within all sequences using BUSCO[18] version 4.1.4. BUSCO was run using default parameters in genome mode with the supplied arthropod sequence database, containing a total of 1066 orthologs from the arthropoda_odb10 database. Taxonomic interrogation was performed by using blobtools[30] version 1.1.1, the *P. virginalis* Petshop 1 WGS dataset[8], and the blast nucleotide database as a hit database. Blobtools was run according to the provided workflow by first creating a BlobDB database using standard parameters, followed by visualization.

***Procambarus fallax* whole-genome sequencing**. Illumina libraries for 28 specimens were prepared by the DKFZ Genomics and Proteomics Core Facility using standard procedures and sequenced on an Illumina HiSeq X platform (PE150 protocol). Raw reads were trimmed and filtered using Trimmomatic[31] version 0.32.

**Whole-genome sequencing data analysis**. Trimmed and processed reads were mapped on the *P. virginalis* mitochondrial genome sequence (Genbank KT074364.1) or on the V1.0 genome assembly using bowtie2[32] version 2.1.0. Alignment files were converted to BAM format and duplicates were removed with SAMtools[33], version 1.9. Furthermore, alignment files were filtered for reference sequences longer than 10 kbp. Multi-sample SNV profiles were calculated for batches of *P. fallax* animals using freebayes (https://arxiv.org/abs/1207.3907, version v0.9.21-7-g7dd41db) with a ploidy setting for diploid organisms. The obtained variant sites from vcf files were subjected to linkage pruning using PLINK[34] version 1.9 and specific parameter settings for windows (50 Kb), window step sizes (10 bp), and $R^2$ threshold (>0.1). The resulting set of variants was used for a Principal Component Analysis (PCA) using default parameter settings in PLINK and plotted using the ggplot2 package in R (version 3.2.1). Alignment files for the mitochondrial genomes were used for consensus calling by the bcftools package from SAMtools. The obtained consensus sequences of *P. fallax* mitochondrial genomes together with the *P. virginalis* reference mitochondrial genome were aligned using Clustal Omega with default parameters. For the resulting multiple sequence alignment, a phylogenetic tree was constructed by the maximum likelihood method implemented in PhyML[35] (v3.1/3.0). The tree in Newick format was visualised via the Interactive Tree of Life online tool[36], with statistical branch support assessed by the bootstrap method[37].

To detect parental nuclear alleles of *P. virginalis*, SNV profiles from all *P. fallax* animals were compared with the set of marbled crayfish-specific heterozygous positions. A matrix was built for each marbled crayfish allele (majority and minority allele) containing either 0 or 1, representing either the presence or absence of the respective alleles in *P. fallax* animals. A neighbor-joining distance matrix was calculated for each allele and an unrooted tree was plotted using APE[38] version 5.3. To identify triploid genomes, biallelic heterozygous variants were extracted from SNV profiles. Focusing only on reference alleles, homozygous single nucleotide polymorphisms in *P. fallax* were discarded. For each variant position, alternative allele frequencies were calculated as the number of alternative allele read observations divided by the total read depth. Finally, frequencies were plotted using the histogram routine in R (version 3.6.1).

**Statistics and reproducibility**. All statistical methods and visualizations were performed using publicly available tools and packages from the statistical computing framework R. Individual packages, parameter settings, and version numbers are mentioned in the respective sections. Linkage pruning was performed using PLINK[34] (v1.9) and an internal $r^2$ statistic threshold of >0.1 for squared correlation of raw inter-variant allele counts. Principal component analysis was performed using the default internal R function with standard parameters. Phylogenetic tree reconstruction was performed by the maximum likelihood method implemented in PhyML[35] (v3.1/3.0) with default parameters. Here, statistical support for each branch was assessed using the bootstrap method[37]. Histogram plots for heterozygous allele frequencies were generated using the default internal R (v3.6.1) histogram function.

This study includes data for PCR genotyping of $N = 92$ *P. fallax* specimens. For each animal, the total number of genetic polymorphisms on 3 gene fragments was extracted and plotted on a scale from 0 (minimum) to 21 (maximum) using the internal R (v3.5.0) heatmap function with default parameters. Additionally, data from $N = 4$ specimens of *P. alleni* were sequenced and integrated. Replicates were defined as a group of individuals with distinct animals from one species (i.e., $N = 92$ for *P. fallax* and $N = 4$ for *P. alleni*). Analysis of whole-genome sequencing data was performed on $N = 28$ *P. fallax* animals collected from 23 independent collection sites in Florida and southern Georgia. Additionally, $N = 2$ publicly available WGS datasets of *P. virginalis* samples were considered for this study. Here, replicates were also defined as the number of individual animals from one species.

**Reporting summary**. Further information on research design is available in the Nature Research Reporting Summary linked to this article.

## Data availability
All sequencing data have been deposited as an NCBI BioProject (accession number PRJNA587442).

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

## Acknowledgements

We thank David Essian, Shawn Clem, Mike Cheek, Noam Lyko, Robert Mattson, Tom Levy, Amir Sagi, Tiffani Manteuffel-Ross, and Shawn Kelly for help with procuring crayfish samples. We also thank Georgios Nikolis and Günter Raddatz for computational support, Nina Glaser for the preparation of PacBio libraries, and Vitor Coutinho Carneiro and Sina Tönges for help with PCRs.

## Author contributions

JG, OM, and HH carried out computational analyses. KH carried out experiments. PP contributed to the conception and design of the analyses. SW provided essential sequencing and computational infrastructure. CES and NJD carried out fieldwork. FL conceived the study and wrote the paper with NJD and additional contributions from JG and NJD. All authors read and approved the final paper.

## Funding

## Competing interests

The authors declare no competing interests.
