## [Peer Review File · Communications Biology]

Reviewers' comments:

Reviewer #1 (Remarks to the Author):

This manuscript describes population genomic analyses to study the origins of the invasive parthenogenetic crayfish, *P. virginalis*. Illumina sequencing of *P. fallax* populations were mapped onto a new PacBio assembly of *P. virginalis*, and compared in several ways to identify the population of origin that *P. virginalis* has evolved from. The body of the paper is well written and explains the study and results clearly. Therefore I have only minor comments.

The methods section was presumably abbreviated for a shorter format journal. I read the supplementary methods for PacBio and annotation and some of that could be put back into the main ms. I would also put Fig S3 back in, since it is strongly relevant to the results (though it does not show *P. virginalis*, which should probably be added).

Some of the other methods are also not completely clear, for example, Fig 1C - how was this tree made? Was this the NJ tree plotted in ape, because that's not really a phylogeny per se...it is a cluster diagram. That is fine for the haplotypes in Fig S4 but should really be a phylogeny here.

Minor:

"However, both animals showed >100,000 homozygous single nucleotide variants from the *P. virginalis* reference genome (Fig. 2C), which demonstrates that they were not *P. virginalis*."
- I don't understand this sentence. Please elaborate a bit.

Reviewer #2 (Remarks to the Author):

In the paper, "Phylogeographic reconstruction of the marbled crayfish origin", Gutekunst et al. conclude that *Procambarus virginalis* originated from the Everglades subpopulation of *P. fallax*. This is a novel finding that can be of interest to carcinologists and a broader community of researchers because it touches on topics related to -omics, invasive species, and phylogeography. I think the content is worthy of publication in *Communications Biology*, but the authors need to provide a better justification for the importance of this study. They also need to improve the clarity of the methods and potentially temper some claims.

Some recommendations to improve the manuscript are below:

Overall Comments:

While the introduction is sound and well-written, I see little in-text justification for the importance of this study. I feel the introduction could be improved by a sentence or two explaining "why this study is needed" and how this can inform or guide future aquarium trade/invasive species research. For example, the abstract touches on the "global invasive" nature of this species and I have read about the destruction this species can have on non-native habitats. Maybe some reference to this is needed in the introduction. I feel the same is needed for the discussion. I am familiar with the crayfish so I immediately found it interesting, however I am not sure others, that know much less about the origin and species, would be convinced by the background and discussion provided.

I am adding this comment here and below, because I do feel it is important for this study to be published. All material sequenced should have a voucher number and linked to curated zoological collection for downstream reference if ever needed.

On page 3 this sentence is confusing to me: "The remarkable global monoclonality can be explained by a recent founder event involving a single animal." Founder events are not mentioned again in the entire manuscript. Are the authors suggesting their study suggest this or was this suggested by previous literature? There is no citation, so I assume they are referring to this study? Please clarify

It is not clear where the *P. virginalis* "reference sequence" came from? Was this generated by the author or from NCBI? This is mentioned on page 3, but no reference is given? Provide what reference is being used for *virginalis*. How valid is this reference material? Details are missing for this part of the manuscript. In the supp materials the authors also do not provide any information on the collection of *virginalis*. Please clarify what references were used for each analysis and how these animals were collected?

The authors use the term "foundational" *P. virginalis* and a definition on how they are defining that term is needed.

Is 2 of 28 relatively rare in other populations with similar triploidy? The authors state this but there is no reference/citation. I am just cautious that limited sampling (n=28) may have been the reason for not finding more triploid individuals. The entire conclusion is based on this finding so adding confidence here is needed.

Lastly, I am curious of the proposed origin of the *P. virginalis* in the aquarium trade. The authors suggest *virginalis* is a direct descendant of *P. fallax* triploids, but how did it then get into the aquarium trade?

Methods:

Overall, I think the methods need a bit more details in the main text. I understand word limitations, but a lot can be gained with little additions. Some recommendations are below.

Add in the specifics on the native range of *P. fallax* so that the reader does not need to refer to S1 unless they want specifics. The addition of the FL range would add little in terms of text, and it is really important when thinking about experimental design.

Which genetic markers, just add the names?

I would require that all animals sequencing in this study have a unique voucher number and be trackable in a zoological collection for downstream accessibility.

I did not see a reference to permits required for collection, especially for this in the National Park. Please provide permitting information.

I was a little confused on the numbering associated with the specimens. Those that have the same numbers were collected from the exact same location? I see Table S2 that shows some of this information but clarification should be in the main manuscript text as these numbers are used over and over in the figures. I now understand the number "28" that is referenced but it is confusing because not figures say "23" with some ".a" or ".b"

On page 5, the authors state *P. fallax* "shares important genetic features with *P. virginalis* and likely represents the foundational population of *P. virginalis*", but in other sections they state it as fact (ex. abstract/discussion). I would like the authors to acknowledge additional future studies/directions are needed to validate these findings and temper statements in other sections.

Figure 1:

Clarify what the numbers (1-23) mean in each of the A, B, and C graphics.

Supplemental Figures:

All colors need to be added in the legend or the figure caption for each independent figure. DO the colors match those in S1.

Reviewer #1 (Remarks to the Author):

1. This manuscript describes population genomic analyses to study the origins of the invasive parthenogenetic crayfish, *P. virginalis*. Illumina sequencing of *P. fallax* populations were mapped onto a new PacBio assembly of *P. virginalis*, and compared in several ways to identify the population of origin that *P. virginalis* has evolved from. The body of the paper is well written and explains the study and results clearly. Therefore I have only minor comments.

>> We thank the reviewer for these encouraging comments.

2. The methods section was presumably abbreviated for a shorter format journal. I read the supplementary methods for PacBio and annotation and some of that could be put back into the main ms. I would also put Fig S3 back in, since it is strongly relevant to the results (though it does not show *P. virginalis*, which should probably be added).

>> Supplementary methods for PacBio sequencing, assembly and annotation were put back into the main manuscript. The previous Fig. S3 is now main Fig. 2, and we have added a new PCA that also contains *P. virginalis* (new Fig. S3). Compared to Fig. 2, this PCA has a considerably lower classification/clusterization performance, which is due to the clonality of the *P. virginalis* genomes and the resulting low number of SNVs between the marbled crayfish sample and the reference genome sequence (also mentioned in the legend to Fig.S3). Nevertheless, Fig. S3 now clearly shows the close relationship between *P. virginalis* and the Everglades population of *P. fallax*.

3. Some of the other methods are also not completely clear, for example, Fig 1C - how was this tree made? Was this the NJ tree plotted in ape, because that's not really a phylogeny per se...it is a cluster diagram. That is fine for the haplotypes in Fig S4 but should really be a phylogeny here.

>> Our tree in Fig. 1C is based on phylogeny after multiple sequence alignment of mitochondrial genome consensus. This has now been clarified in the Methods section. We now also indicated in the figure legend that the bootstrap values for the main branches are 100%.

4. "However, both animals showed >100,000 homozygous single nucleotide variants from the *P. virginalis* reference genome (Fig. 2C), which demonstrates that they were not *P. virginalis*." - I don't understand this sentence. Please elaborate a bit.

>> We have clarified this in the text: "However, both animals also showed a considerable number of genetic polymorphisms (>100,000 homozygous single nucleotide variants) towards the *P. virginalis* reference genome (Fig. 2C). This is more than an order of magnitude above what has been described for *P. virginalis* (ref. 5) and demonstrates that these animals are genetically distinct from *P. virginalis*."

Reviewer #2 (Remarks to the Author):

1. While the introduction is sound and well-written, I see little in-text justification for the importance of this study. I feel the introduction could be improved by a sentence or two explaining "why this study is needed" and how this can inform or guide future aquarium trade/invasive species research. For example, the abstract touches on the "global invasive" nature of this species and I have read about the destruction this species can have on non-native habitats. Maybe some reference to this is needed in the introduction. I feel the same is needed for the discussion. I am familiar with the crayfish so I immediately found it interesting, however I am not sure others, that know much less about the origin and species, would be convinced by the background and discussion provided.

>> Marbled crayfish are not known to be "destructive". However, we have now added two paragraphs of text to the introduction to better explain the various points that were raised by

the reviewer, including the importance of the study. We have also revised and expanded the abstract correspondingly.

2. I am adding this comment here and below, because I do feel it is important for this study to be published. All material sequenced should have a voucher number and linked to curated zoological collection for downstream reference if ever needed.

>> See point 10 below.

3. On page 3 this sentence is confusing to me: "The remarkable global monoclonality can be explained by a recent founder event involving a single animal." Founder events are not mentioned again in the entire manuscript. Are the authors suggesting their study suggest this or was this suggested by previous literature? There is no citation, so I assume they are referring to this study? Please clarify

>> Our revised introduction better explains the foundational event: Comparative genome sequencing of numerous specimens from diverse sources and locations confirmed that the global *P. virginalis* population is largely monoclonal with only very limited genetic differences between geographically separated populations (refs. 2, 5, 8). This monoclonality can be reasonably explained by a recent founder event involving a single animal. The identity of this foundational animal has remained largely unknown.

4. It is not clear where the *P. virginalis* "reference sequence" came from? Was this generated by the author or from NCBI? This is mentioned on page 3, but no reference is given? Provide what reference is being used for *virginalis*. How valid is this reference material? Details are missing for this part of the manuscript. In the supp materials the authors also do not provide any information on the collection of *virginalis*. Please clarify what references were used for each analysis and how these animals were collected?

>> This has now been clarified: "... every analyzed *P. fallax* had at least one polymorphism (Fig. 1A) with the published *P. virginalis* reference sequence (ref. 8)." Our revised introduction also explains the generation of the *P. virginalis* reference sequence.

5. The authors use the term "foundational" *P. virginalis* and a definition on how they are defining that term is needed.

>> See point 3 above.

6. Is 2 of 28 relatively rare in other populations with similar triploidy? The authors state this but there is no reference/citation. I am just cautious that limited sampling (n=28) may have been the reason for not finding more triploid individuals. The entire conclusion is based on this finding so adding confidence here is needed.

>> We have added the following sentence in the second paragraph of the discussion: "In comparison, triploid parthenogenetic individuals in populations of the New Zealand mud snail (*Potamopyrgus antipodarum*) were substantially more frequent (53%, N=860) (ref. 22)." In addition, we are now also emphasizing that future studies will be required to further validate our findings, see point 13 below.

7. Lastly, I am curious of the proposed origin of the *P. virginalis* in the aquarium trade. The authors suggest *virginalis* is a direct descendant of *P. fallax* triploids, but how did it then get into the aquarium trade?

This is now clarified and referenced in the introduction: "The animals were originally introduced into Germany in 1995, as an ornamental aquarium pet from the USA (ref. 2), and then became rapidly distributed through the aquarium trade (refs. 1, 3)."

8. Add in the specifics on the native range of *P. fallax* so that the reader does not need to refer to S1 unless they want specifics. The addition of the FL range would add little in terms of text, and it is really important when thinking about experimental design.

>> This has now been clarified in our revised introduction: "*Procambarus fallax* are sexually reproducing diploids that are endemic to peninsular Florida and southern Georgia (ref. 17)."

9. Which genetic markers, just add the names?

>> Clarified in the text: For genotyping, we used partial sequences of the mitochondrial *cytochrome b* (*CytB*), nuclear *DNA methyltransferase 1* (*Dnmt1*) and mitochondrial *cytochrome c oxidase 1* (*Cox1*) genes (see Methods for details).

10. I would require that all animals sequencing in this study have a unique voucher number and be trackable in a zoological collection for downstream accessibility.

>> Unfortunately, this was not possible. While PCR genotyping can be done from tissue biopsies, more comprehensive analyses (whole-genome sequencing, PacBio sequencing, whole-genome bisulfite sequencing) require substantially larger amounts of DNA that could only be obtained from whole animals.

11. I did not see a reference to permits required for collection, especially for this in the National Park. Please provide permitting information.

>> Georgia Department of Natural Resources Scientific Collection Permit 115621108 and State of Florida Collection Permits S-19-10, S-20-04 are now indicated in the Methods section. It should be noted that *P. fallax* is not a protected species in Florida and can be collected as a fishing bait. Also, no *P. fallax* were collected from Everglades National Park (the Everglades wetland is an ecosystem much larger than just the park in the most southern portions of the state).

12. I was a little confused on the numbering associated with the specimens. Those that have the same numbers were collected from the exact same location? I see Table S2 that shows some of this information but clarification should be in the main manuscript text as these numbers are used over and over in the figures. I now understand the number "28" that is referenced but it is confusing because most figures say "23" with some ".a" or ".b"

>> This is now clarified in the text (and in the legend to Fig. 1): "Samples were numbered by their sampling locations from 1 (northernmost) to 23 (southernmost)". Further down, we write: "For a phylogeographic analysis of the *P. fallax* population, we selected one representative specimen from each location (numbered 1-23) for whole-genome sequencing (Tab. S2). For five locations, where the PCR results had suggested a greater level of genetic heterogeneity, two specimens were selected (labelled "a" and "b")."

13. On page 5, the authors state *P. fallax* "shares important genetic features with *P. virginalis* and likely represents the foundational population of *P. virginalis*", but in other sections they state it as fact (ex. abstract/discussion). I would like the authors to acknowledge additional future studies/directions are needed to validate these findings and temper statements in other sections.

>> We agree with the reviewer and have tempered the corresponding statements. Also, we have added two sentences at the end of the first paragraph in the discussion to outline future studies that could be done to validate our findings.

14. Figure 1: Clarify what the numbers (1-23) mean in each of the A, B, and C graphics.

>> This has now been clarified in the figure legend: Sample numbers indicate individual sampling locations from 1 (northernmost) to 23 (southernmost); a and b indicates independent animals collected from the same location.

15. Supplemental Figures: All colors need to be added in the legend or the figure caption for each independent figure. DO the colors match those in S1.

>> The colors in Fig. S1 now precisely match match the "standard" colors for the subpopulations. Also, subpopulation colors are now explained in each supplemental figure legend.